# Can We Steer Nursing Home Residents Away from Loneliness? A Qualitative Study of Cycling without Age

**DOI:** 10.3390/geriatrics9040088

**Published:** 2024-06-25

**Authors:** Sara Kruse Lange, Maiken Hauge Stig, Martin Eghøj, Christina Bjørk Petersen

**Affiliations:** 1The Department of Sport Science and Clinical Biomechanics, University of Southern Denmark, 5230 Odense, Denmark; 2National Institute of Public Health, University of Southern Denmark, 1455 Copenhagen, Denmark; masti@sdu.dk (M.H.S.); maeg@sdu.dk (M.E.)

**Keywords:** loneliness, qualitative methods, nursing home, mental health, bicycling, Cycling Without Age

## Abstract

Background: Loneliness among nursing home residents is an increasing public health issue and consists of a combination of social, emotional, and existential loneliness. Cycling Without Age (CWA) involves taking nursing home residents on trishaw rides pedaled by trained volunteer ‘pilots’. This study aims to explore nursing home residents’ lived experiences of CWA and whether participation in CWA can mitigate experiences of loneliness. Methods: A qualitative phenomenological design was used. We conducted three observations and eight interviews: semi-structured interviews (n = 5) and informal interviews (n = 3) with passengers in CWA. Data were analyzed using reflexive thematic analysis. Results: Three themes were developed: 1. creating meaningful communities (related to the social mechanism connected to participating in CWA), 2. breaking the monotony of everyday life (related to how the passengers experience CWA as a meaningful activity), and 3. reconnecting to oneself (related to the meaningful experience the passengers have when they are connected to their local communities and reminiscence). Conclusion: Taking part in CWA may mitigate loneliness, as passengers perceive it as being meaningful. These results strengthen the notion that participating in meaningful activities hold the potential to mitigate feelings of loneliness among nursing home residents.

## 1. Introduction

The proportion of elderly who experience loneliness is high and has been increasing in Denmark in the past few decades [1,2,3]. More than half of all nursing home residents experience loneliness [4,5] possibly due to age-related changes, personal losses, separation from friends and family, unfulfilled needs for meaningful relationships, and institutionalization [6].

Loneliness is a multidimensional experience and consists of social, emotional, and existential loneliness [7,8]. Mansfield et al. [7] conceptualize social loneliness as an objective condition framed by numbers of social connections, a subjective feeling of being isolated, and lacking access to a satisfying social network; emotional loneliness as social isolation, a loss or lack of good quality social relationships, and a lack of a sense of belonging or recognition; and finally, existential loneliness as a fundamental separateness from others and feeling cut off from the outside world, not simply as the absence of meaningful relationships and negative emotional experiences. According to Jansson et al. [8] elderly living in long-term-care facilities may experience social loneliness because they lack company or networks and support from peers or feel lonely in a crowd. Emotional loneliness stems from meaninglessness in life, feeling all alone and longing for meaningful others. Finally, existential loneliness is characterized by experiencing emptiness and feeling fundamentally left behind [8,9]. Across all the dimensions, loneliness is described as an unpleasant and unwanted emotional experience [10].

It is well documented that loneliness is associated with lower life satisfaction [11], poorer physical and mental health [12,13], and increased mortality [14]. Furthermore, research has shown that experiences of reduced meaning in life are associated with feeling lonely [15]. Thus, having meaningful experiences can be considered as a way to mitigate the feeling of loneliness.

Several interventions have been developed to reduce loneliness among elderly living in nursing homes [16,17]. However, most of these include therapies and/or leisure/skill development [16]. Cycling Without Age (CWA) is a program that takes nursing home residents, called passengers, on trishaw rides piloted by volunteers in the local community. CWA started as a social non-profit organization in Denmark in 2012 and is now implemented in more than thirty countries globally. The organization is driven by volunteers and donations from private foundations, courses, members, and municipal funds. CWA is available in 45 out of 98 municipalities in Denmark. The selection of passengers for rides is usually a collaborative process between the pilot and an activity coordinator from the nursing homes. This ensures that different residents get to go on a ride [18].

Although CWA is widely implemented around the world, little evaluation exists on how it affects the passengers’ experience of loneliness. Previous findings suggest that participating in CWA rides has positive short-term effects on the participants’ mood and wellbeing [19] and is associated with of positive emotions [20]. Moreover, participation in the bike rides facilitate socialization among residents [21] and provide an opportunity for them to spend time outside the nursing homes, rediscovering familiar places within the community [22].

This study explores the experiences of nursing home residents who participate in Cycling Without Age (CWA) rides, with a particular focus on their perceptions of loneliness. It aims to understand the residents’ lived experiences of the rides and how they influence feelings of loneliness. The findings from this study will contribute to evidence-based recommendations for enhancing the quality of life for this vulnerable population.

## 2. Materials and Methods

### 2.1. Study Design

This is a qualitative phenomenological study using reflexive thematic analysis. We conducted three observations and eight interviews: semi-structured interviews (n = 5) and informal interviews during observations (n = 3) with passengers in CWA. It is essential to note that this study is an integral component of a larger evaluation initiative of CWA titled “When movement moves” [23]. The term ‘passengers’ will be employed to refer to the nursing home resident participants involved in this study.

### 2.2. Setting and Data Collection

Interviews took place in the passengers’ apartments at four different nursing homes. In three of these nursing homes observations were conducted following pilots and passengers during CWA rides. Data were collected over a two-month study period (February–March 2022). During this period, the Danish Health Authority’s guidelines on COVID-19 were still in effect. This meant that face masks were required while moving around the nursing homes [24]. All the passengers articulated a strong wish to complete the interviews without face masks, which we complied with.

### 2.3. Participants

A purposive sampling technique was used to recruit passengers (n = 8) [25]. The initial contact with potential participants was facilitated by activity coordinators from nursing homes and pilots associated with CWA, who acted as gatekeepers (n = 7). We provided them with information about the study’s purpose and inclusion criteria. To be eligible, passengers had to reside in a Danish nursing home affiliated with the CWA community. Eligibility was assessed mainly by the gatekeepers, in cooperation with the project team. Passengers deemed unable to participate in a minimum 20-min interview were excluded from the interview phase but were still considered for observations. According to the gatekeepers, it was challenging to find passengers who met the inclusion criteria, as many of the passengers were unable to participate in an interview because of dementia, the loss of the ability to speak, or other health challenges. One gatekeeper was unable to identify any eligible passengers. Another identified one passenger across five nursing homes. Additionally, comprehensive COVID-19 restrictions limited the access to nursing homes and, consequently, impeded interactions with potential participants.

Ten nursing homes were contacted, but we were only able to include passengers from four nursing homes. Given the abovementioned criteria and constraints, we invited ten passengers, of whom eight accepted the invitation to participate (Figure 1). This reflects the significant health challenges faced by many elderly individuals in nursing homes. We obtained informed written consent from all the participants. Pseudonyms for the passengers are used throughout the paper.

Five passengers were interviewed in their apartments (Anja, Carsten, Bent, Elly, and Frida), while three (Edith, Timmy, and Andy) were informally interviewed during observations. Information on selected health condition of the participants was obtained from the staff. Table 1 shows the characteristics of the participants. One informant with from very mild to mild dementia and one informant with Broca’s aphasia were included. 

According to Kirkevold and Bergland [26] an interview sample that does not include frailer informants may result in an incomplete picture of a given phenomenon. For this reason, to increase the quality of our data, we did not exclusively include experiences and views from passengers in good health and high cognitive function but also from passengers with mild cognitive and communicative limitations who were to participate in a 20-min interview. However, based on ethical concerns, the inclusion criteria was not to be demented, have a verbal language, able to provide first-hand data and consent to participate on an informed basis. Our observations, however, included non-verbal passengers, and this provided us with data from non-elite residents. 

To make the passengers feel comfortable during interviews, we allowed silences and waited if someone struggled to find the right words. We did not correct the wrong use of names or words but rather asked passengers to elaborate further to avoid misunderstandings [26]. In addition, our choice to include observations helped us to provide a voice for the residents who had difficulty expressing themselves verbally. 

### 2.4. Interviews

Interviews were conducted by MHS and SKL. We used a semi-structured interview guide [27] consisting of three themes: (1) experiences of participating in CWA, (2) experiences of social network, and (3) experiences of loneliness. To develop our semi-structured interview guide, we used the framework proposed by Kallio et al. [27]. In the development, we went through the following steps: (1) identifying the prerequisites for using semi-structured interviews, (2) retrieving and using previous knowledge, (3) formulating the preliminary semi-structured interview guide, (4) pilot testing the interview guide, and (5) presenting the complete semi-structured interview (Appendix A). This guide helped to prioritize question sequencing and eased passengers into discussions, particularly crucial when addressing a sensitive topic, like loneliness, where we addressed the theme about loneliness by making the vignettes “*We know in advance that elderly people who move into nursing homes often feel lonelier compared to other times in their lives. Why do you think that is?*”. Vignettes are useful for addressing sensitive topics, as they allow the interviewers to ask informants to reflect on a scenario rather than diving directly into their own experiences [25].

We used open-ended and follow-up questions, and active listening, to encourage the spontaneous and in-depth sharing of personal feelings and stories. Probing questions were used to clarify or expand upon responses [27]. 

The interviews, lasting between 20 and 45 min, were recorded and transcribed. We made a conscious attempt to ensure the comfort of the informants during the interviews, and to not leave them agitated or upset at the conclusion of the interviews, by asking how they were feeling and how they felt about the interview. Furthermore, we informed the nursing home staff about the topic and the agenda of the interviews and asked whether they would be attentive to delayed reactions. To the best of our knowledge, no passengers were affected by the interviews in any negative way. Because of the sensitive and stigmatized nature of the loneliness topic, emotional reactions and observed mood shifts were noted in the transcripts.

### 2.5. Observations and Informal Interviews

We actively participated in two one-hour-long rides as passengers during observations beginning 15 min prior to and after the ride. Our approach focused on allowing passengers and pilots to initiate conversations. The third observation occurred at the nursing home and involved three informal passenger interviews. Detailed field and reflexive notes were promptly recorded or dictated following each observation, following the methodology outlined by Green and Thorogood [25] (Appendix A). The aims were to understand the passengers’ roles, familiarize ourselves with the context, and gain insights into the CWA participation experience.

### 2.6. Data Analysis

We used the six phases of Braun and Clarke’s reflexive thematic analysis to analyze the field notes and the transcribed interviews [28]. Figure 2 shows the coding and thematic development process. Our approach was a bottom-up inductive coding process, where we began by familiarizing ourselves with the data (phase 1). Afterwards, we assigned open, descriptive codes using both semantic and more nuanced latent coding, which summarized the content of each data extract in relation to the research questions (phase 2). To generate the initial themes, we sorted the codes with a common underlying meaning into broader groups (phase 3). The initial themes were reviewed and further developed among the full author group (phase 4). Each theme and sub-theme were given a definition, and the original data were revisited to ensure a proper understanding of what the passengers had expressed. Finally, the themes were adjusted after feedback from the team. This resulted in three final themes in which two themes had associated sub-themes (phases 5 and 6).

We adopted our reflexive approach, as we consider subjectivity as a valuable and key aspect of qualitative sensibility, and we found that it paid off when working with a stigmatized and sensitive topic, such as loneliness [28]. We used the qualitative data software NVivo to code and organize the data.

## 3. Results

We identified three themes: (1) creating meaningful communities, (2) breaking the monotony of everyday life, and (3) reconnecting to oneself. Within these themes, we identified four sub-themes (Table 2).

All the passengers were asked open-ended questions about their experiences with the CWA, including any negative aspects. Some passengers expressed that they preferred not to go on rides if the weather was too cold or rainy, that it could feel cramped to have two passengers on one bike, and that they were dependent on a pilot to offer them a ride. However, despite great attempts to ask about and include any bad or negative sides for participating in CWA, all the passengers were overwhelmingly positive and grateful for the rides, and this is reflected in the three themes. Consequently, three themes do not reflect any negative association or bad experiences.

### 3.1. Creating Meaningful Communities

The theme *Creating meaningful communities* consists of the sub-themes “Meaningful relationships” and “Shared experiences” and covers the social aspects associated with participation in CWA.

#### 3.1.1. Meaningful Relationships

Passengers describe CWA as facilitating relationships between passengers and/or pilots. When passengers go solo, they can more easily interact with the pilot. When two passengers go together, they can interact with each other as well as the pilot. Frida talks about sitting next to another passenger:


*It is tight. But it is so cozy to talk about what we see, so in that way, being two on a bike is nice.*
(Frida)

Sitting with someone is also highlighted by Anja:


*In the beginning, I was riding with another resident. She loved flowers. We talked about all the wildflowers we saw on the country road and which kind they were. She remembered and I remembered. We loved flowers. That was nice. Unfortunately, she is no longer here.*
(Anja)

Both quotations show positive experiences with another passenger, highlighting the social aspects of the rides, despite the cramped physical space. The pilots also play a special role in creating positive social experiences. All the passengers talk about the pilots with gratitude. For instance, Elly says:


*I have been out cycling 4–5 times with the same pilot. It gives a certain sense of security to know the person you are riding with a little. One becomes happy about that…well… it becomes a little more familiar…one gets closer to one another.*
(Elly)

A different view is presented by Bent:


*Well, that was very good. When one is a little curious then it is nice to be told what is happening […] I barely remember what we talked about, but it was an experience to be told about the development in the area. The pilot is good at explaining, one must say, he knew many things about the area. He really made an effort.*
(Bent)

Elly emphasizes a sense of familiarity, while Bent is impressed by how the pilot facilitates the overall experience by explaining and telling stories.

The pilots can also play a positive role by creating a connection between the nursing home and the world outside: ‘*one of my friends is one of the pilots*’, as Timmy explains during an observation and sheds light on how the rides provide a possibility for him to maintain a friendship with someone outside the nursing home.


*All the passengers explicitly, but differently, say that they appreciate the effort and time the pilots put into it. While Bent exclaims: ‘that is really something!’.*
(Observational notes)

Frida articulates: 


*I think it is a great thing that someone wants to cycle with us, I really do […] You know we are lonely here, so we welcome the bike rides with open arms.*
(Frida)

#### 3.1.2. Shared Experiences

The bike rides create a shared experience for the passengers and pilots. They facilitate conversation, comfortable quiet spells, and social connections. This occurs even if the passengers or the passengers and pilots meet for the first time.

Bent talks about the first time he and the pilot rode together in the following:


*The pilot is telling us about an old factory that has been demolished in a place where new rentals will be built. The pilot continuously talks about where we are. When we bike through a forest area, Bent and the pilot talk about the trees that are partly cut down.*
(observational note)

Beyond facilitating conversation, the passengers talk about quiet spells during rides and describe them as comfortable. The rides go through the landscape; the scenery changes, and observing it allows passengers to ‘*just enjoy being together without talking*’, as Carsten says during the interview. Carsten also shows a photograph and says: ‘*The guy in the photo; he can’t speak. But it is nice when we are out on a bike ride. We just enjoy being together without talking*’.

These quotes show how shared experiences on rides spark conversation and comfortable quiet spells, which contribute to positive and comfortable experiences.

Further, the shared experiences also often include coffee breaks:


*We [the passengers] talk a lot together about the bike rides, about where we are going and what we are seeing. There are also two or three from other parts of the nursing home whom we also talk with when they are out on the bikes with us. We also talk during the coffee breaks on the bike rides. We sit next to one another in a circle, and we all talk. It is social in that way.*
(Anja)

When the passengers talk about the coffee breaks, they emphasize talking and laughing in a cozy atmosphere. We observed this firsthand: 


*The coffee break was cozy. We talked and laughed, and one of the passengers talked about a place where they sometimes got ice cream’.*
(observational notes)

After the rides, the passengers often talk about their experiences with friends, other residents, family/visitors, and nursing staff, as mentioned by Bent: ‘*I told the nursing staff, and some of the others that live here [nursing home residents], that it [a bike ride] was fun and a different experience*’.

Elly also offers insight into how a bike ride can facilitate conversations about the trip:


*Well, you come home with new inputs… And that is something that we talk about with one another…If one of the other girls [friends in the nursing home] also has been on a bike ride, then we discuss where we have been.*
(Elly)

For Bent, Elly, and the other passengers, the rides are not self-contained events; in fact, they spark conversations about the rides, giving shared experiences and new inputs for conversations.

### 3.2. Breaking the Monotony of Everyday Life

Because the bike rides break the monotony of life in general and the activities that nursing homes provide, the rides are meaningful to the passengers. Anja experiences the difference between going on the rides and other activities in the nursing home:


*Well, I feel good. I like it [the ride]. Now, I can manage a little again until my next trip […]. But you can’t get out [of the nursing home] as much. You must accept staying at the nursing home. I do have some volunteering companions, who come every week and then we go for a little walk outside. But once I’ve been on a ride, then I can manage. Then, after a little while, one hopes one gets to go on a bike ride again soon. I look forward to it [the bike ride].*
(Anja)

Elly also gives her take on what a ride provides:


*Well, it gives you a fresh input into everyday life; to get out and experience something because it is not much we experience inside these four walls.*
(Elly)

For Carsten too, the rides are invigorating: 


*I have also seen a fallow deer on the field and how they run. Yes, I see many things, I think, because suddenly something happens […] When we bike towards the hill, there are plums, and even if I can’t reach them, I feel like trying. I see the leaves falling slowly down. It is so beautiful! [He speaks more rapidly, smiles more broadly, and gets tears in his eyes.] I think it is lovely to bike.*
(Carsten)

The quote above highlights how some of the passengers experience the rides differently from other activities and from the monotony of everyday life at the nursing home. However, although Anja highlights getting renewed energy and vigor after a ride, Elly emphasizes the fresh inputs a ride can provide as a contrast to her everyday life, while Carsten focuses on the experience of being out in nature and following the seasons.

How the rides deviate from everyday life in the nursing homes becomes apparent in the passengers’ expressions. During an informal interview, Edith says:


*It is limited what happens in such a nursing home, where many are not quite well.*
(Elly)

Anja adds a more nuanced perspective, elaborating on how living in a nursing home, in general, can be lonely:


*I would think that loneliness is if you get up in the morning and think: “What should you do today? How will the day go? Are you just going for a bit of dinner and back again? Are you just sitting on the sofa and watching television? Is someone coming to talk to you today? There probably isn’t.” I believe that is when you feel a little lonely, a little left behind.*
(Anja)

In this quote, Anja describes how one can feel left behind in the nursing home contrasts the feelings invigorated after the rides.

### 3.3. Reconnection to Oneself

The final theme, *Reconnection to oneself* consists of the sub-themes “To revisit and reminisce” and “To be seen and noticed”. This theme explores how participation in CWA facilitates reconnecting to themselves and to their old neighborhoods, which causes them to reminisce.

#### 3.3.1. *To Be Seen and Noticed*

To be seen and noticed outside the nursing home is a topic commonly brought up when passengers talk about CWA. Frida recalls a ride:


*It is nice to sit on the bike and be able to see everything one wants. Perhaps one is even lucky enough to meet someone one knows and can wave to. And the kids around, they look at me on the bike [smiles broadly, and her face brightens up].*
(Frida)

During our observation with Bent, we passed three schoolchildren:


*They smiled and looked inquisitively at Bent and the bike. Bent looked at them with a broad smile on his face. Bent continued to have a broad smile on his face for a while after we passed by them.*
(observational notes)

Both Bent and Frida seemed to enjoy being seen and noticed by others. Bent highlights the schoolchildren passing by, and Frida highlights someone waving at her and the children noticing her. This shows how the rides can offer passengers a connection to the outside world and how this connection brings joy. 

#### 3.3.2. *To Revisit and Reminisce*

The passengers describe how they appreciate reminiscing when visiting familiar places. Anja describes the rides going to places of her past as outstanding experiences and fundamentally meaningful:


*I was born and raised not too far from this city, so I know it all. But to see it all again is the joy of reminiscing. I used to bike myself, but when I ended up here [in the nursing home] one thought: “You will never get out there anymore.” YES! Then all of a sudden, we take a trip on the bike out there. That, I would never have dreamt that I got to see it once more. THEN you become happy! [Anja is very moved. Her voice breaks, and her eyes are tearful.] Then, one thinks that you are not totally gone yet. You still get to come out and see some of the old.*
(Anja)

Elly had a specific ride in mind:


*We stopped, and the pilot told me that his wife worked up there. He called her and then she stood by the window and waved at us…I used to work in that same building. That was really something.*
(Elly)

Both quotes show how CWA can create meaningfulness and existential reflections about life. Anja explains how the experience moved her existentially, while Elly emphasizes how the experience made a significant impact on her. The existential perspective also appears during the interviews, as the passengers mention their own mortality, e.g., when Bent adds, ‘I will not be here forever.’ during his interview, or when Carsten says, ‘Whilst I am still here’, or when Anja mentions that ‘We will all die at some point.’ These existential reflections show that trips to familiar places and remembering the time before moving to a nursing home are important to the passengers.

Visiting places that the passengers have a connection to and deciding where to go enhance the passengers’ positive associations. Frida said:


*...then we talk about where we would like to go. I want to go near the stream. It is so lovely [Her voice becomes more joyful.]. There is also a forest kindergarten. It is fun to watch, and people are going for walks. […]. It is a really good thing.*
(Frida)

The passengers describe how the bike trips gives them a feeling of freedom, independence and autonomy which contrasts with the feelings of limitation in the everyday life at the nursing home. Bent describes: ‘*I can’t go out and buy anything. The shops are too far away*’. Independence as a passenger vastly contrasts with the lack of independence as a nursing home resident.

Finally, CWA is presented as an activity the passengers are proud off, like Carsten, who points to a picture on his desk and adds: ‘*THAT was a great day*’. or Timmy talking about the large, framed photo on his wall: ‘*That’s me on the bike*’. Timmy’s photo is next to photos of him in his old home and other things reflecting what he likes and who he is. The display of the photos of the rides with CWA can be a way for reminiscing about the experiences of being reconnected with one’s old neighborhood and childhood home.

## 4. Discussion

In this study, we set out to explore the lived experiences of passengers participating in CWA, with a specific focus on whether these experiences might mitigate loneliness among nursing home residents. Through qualitative analysis, we identified three themes, which encapsulate the essence of these experiences: (1) creating meaningful communities through social interactions and shared experiences with others, (2) breaking the monotony of the life at the nursing homes, and (3) reconnecting to oneself through being seen and noticed and reminiscing. 

First, the passengers described CWA as an activity that provided meaningfulness in a fundamental and profound way in their everyday lives and gave them a meaningful connection to communities outside the nursing home. This contrasted with the way the passengers described feeling left behind and enclosed in the nursing home. As existential loneliness is characterized by experiencing emptiness and feeling fundamentally left behind [8,9], participating in CWA may mitigate the experience of existential loneliness, as CWA can give the passengers a meaningful and deeper element in their life.

Second, the passengers describe the CWA rides together with others (passenger or pilot) as comfortable and enjoyable as well as creating a sense of emotional connectedness. This supports the notion that participation in CWA can mitigate emotional loneliness, as emotional loneliness is described as feeling all alone and longing for meaningful others [8,9].

Finally, the passengers spoke about how CWA provides them with an opportunity to interact in different contexts, which they would otherwise not be privy to. This highlights how CWA can mitigate social loneliness, as it reduces the negative experiences of missing company or networks or lacking social support from peers [8]. Thus, it seems that all three dimensions of loneliness are mitigated by participating in CWA.

Overall, the themes identified in this study align with findings from similar studies. Cyarto et al. [22] found that CWA was associated with experiencing reminisced positive emotions and gave meaningful experiences. Gray et al. [20] found that the core of CWA was meeting new people and engaging in a shared experience, and McNiel and Westphal [21] found that participating in CWA gave nursing home residents the possibility of fresh air and social connectivity. Our study provides a nuanced and in-depth look at how CWA might mitigate loneliness because it is meaningful and provides social company for the passengers.

According to Mansfield et al. [7], most research on loneliness focuses on social loneliness, and little is known of determinants and interventions in existential loneliness. During the interviews, the passengers spontaneously shared existential thoughts on mortality. In line with this finding, previous studies have shown that existential loneliness is highly prevalent among nursing home residents [29], as they often experience feelings of emptiness (lack of meaning) and loss of identity and social status [8,9]. Additionally, according to Tam and Chan [30], meaningful activities can mitigate the experience of loneliness, which is supported by Macià et al. [15], who find that reduced meaning in life is associated with feelings of loneliness. 

Tam and Chan [30], furthermore, found that the more meaningful the perception of an activity is, the lower the likelihood of feeling lonely. Tam and Chan’s findings support our belief that CWA rides may contribute to mitigate loneliness because the activity is perceived as being meaningful. For this reason, our results make a unique contribution to the field, as we focus on loneliness as a multi-dimensional phenomenon and find that CWA, as an activity, provides value to the passengers, as it affects them fundamentally and meaningfully. According to Smale et al. [31], the sense of belonging to a community is the strongest factor in mitigating loneliness among older adults. Our results indicate that CWA gives the passengers the opportunity to become a part of the world outside. Thus it may facilitate a reconnection to their past and/or local communities, which they are unable to achieve within the nursing home. 

Our findings show that the passengers experience emotional connectedness to other people through conversations, which go beyond the rides. This makes CWA a shared interest, and a topic for conversations in the homes, among the residents and staff, family, and friends, which might contribute to mitigating emotional loneliness. Previous research by, for instance, Kharicha et al. [32], support this by showing that having a shared interest makes it easier for nursing home residents to get involved with each other and that by attending such an activity, a social element develops over time. Similarly, Fessman and Lester [33] found that nursing home residents with close bonds to other residents reported less depression and loneliness. Thus, our study underscores the impact of meaningful activities, like CWA, in enhancing social wellbeing and resilience among older adults in institutional settings. 

However, implementing Cycling Without Age (CWA) on a larger scale may present challenges because of the increasing elderly population, the rising demand for care, and the decreasing number of available caregivers [34]. Moreover, the volunteer model of CWA may not be globally feasible. Nonetheless, as the population of the elderly grows, more individuals who have retired may become available to serve as volunteers. Therefore, despite the limited volunteer availability, the benefits of CWA could still be realized by making it a regular part of care or by integrating CWA activities into nursing home routines. This integration can provide meaningful experiences through resident interactions, outdoor activities, and community connections.

### 4.1. Strengths and Limitations

This study gives valuable insight into the importance of how an activity, such as CWA, can affect nursing home residents’ perceptions of loneliness. The main strength of our study is that data comprise both observations and interviews, which increased our ability to capture nuances in the views of the nursing homes’ residents. Our observational notes gave valuable insight into the setting and helped us to understand the experience of a ride. Additionally, the observations gave a voice to the nursing home residents who had difficulty expressing themselves verbally. We included informants from four different nursing homes to examine common features of rides in different contexts.

A potential limitation is the exclusion of non-CWA residents, preventing us from understanding their experiences of nursing home life or their perceptions of seeing others participate in CWA. Our results are based on informants who are active CWA participants, which may introduce bias and limit the diversity of perspectives in our findings.

Another limitation is the fairly low number of informants included in our sample, which may have restricted the range of perspectives and insights represented in our study. However, despite residents living at homes at different ends of the country, similar experiences emerged across the informants. 

### 4.2. Implications for Practice and Future Research

Our sample is limited to active CWA participants, who may inherently be more social and less lonely to begin with compared to the general nursing home residents. However, because all the passengers described a sense of feeling left behind or, to some degree, felt isolated within the nursing homes, our research emphasizes the importance of activities that create a connection between nursing home residents and their communities. As such, by listening to the voices of nursing home residents, steering away from loneliness might be plausible. Incorporating CWA programs into the activities offered within nursing home settings could provide residents with meaningful opportunities for social engagement and community connection. Moreover, healthcare professionals and caregivers can use our insights to tailor support services and interventions that address the unique needs and preferences of older adults in institutional care settings. By recognizing the importance of meaningful activities in the work for preventing loneliness, practitioners or volunteers can play a pivotal role in enhancing meaningful activities for nursing home residents and fostering a sense of belonging within their communities.

## 5. Conclusions

The literature suggests that meaningfulness is the predominant mitigator of loneliness. Our study gives an in-depth exploration of how passengers experience CWA and insights into how these rides may influence the feeling of loneliness among nursing homes’ residents. Existential loneliness is highly prevalent among nursing home residents. To the best of our knowledge, this is the first study to explore various dimensions of loneliness among nursing homes’ residents and the potential mechanisms of CWA that may influence loneliness. We find that, taking part in CWA may mitigate loneliness, as passengers perceive it as being meaningful. Our results strengthen the notion that participating in meaningful activities, such as CWA, hold the potential to mitigate feelings of loneliness among nursing home residents.

However, more research is required to explore similar activities among nursing home residents, to explore and understand how and why activities can create an experience of meaningfulness, and to promote mental health and wellbeing among nursing home residents. 

## Figures and Tables

**Figure 1 geriatrics-09-00088-f001:**
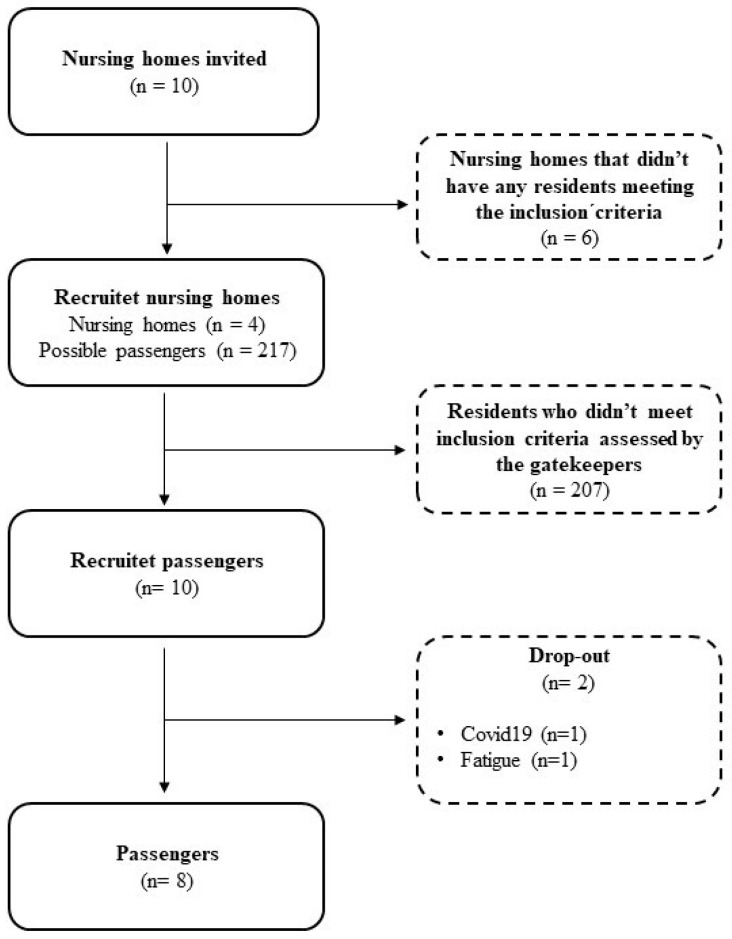
Flowchart showing the recruitment process.

**Figure 2 geriatrics-09-00088-f002:**
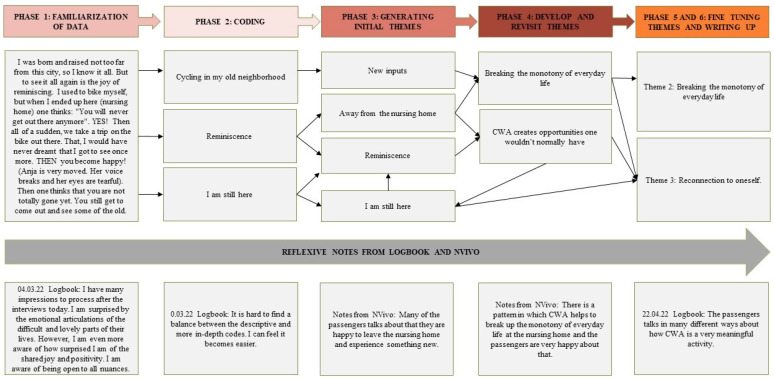
Reflexive thematic analysis process.

**Table 1 geriatrics-09-00088-t001:** Participants’ demographics.

Informant	Sex	Age	Civil Status	Precipitation in CWA	Nursing Home No.	Years in Nursing Home	Cognitive Status *
Interviews							
Anja	Female	89	Widow	>2 years	1	>8 years	No known disorders
Carsten	Male	77	Widower	<1 year	1	<1 year	Aphasia
Bent	Male	90	Relationship at the nursing home	<1 month	2	<1 month	From very mild to mild decline
Elly	Female	82	Widow	>3 years	3	>3 years	No known disorders
Frida	Female	90	Widow	>3 years	3	>3 years	No known disorders
Informal interviews during observations							
Edith	Female	94	Widow	>3 years	4	>8 years	No known disorders
Timmy	Male	78	Relationship at the nursing home	>3 years	4	>2 years	No known disorders
Andy	Male	76	Relationship outside the nursing home	>1 year	4	>6 months	From very mild to mild decline

* Information about cognitive status was provided by the nursing home staff during the recruitment phase.

**Table 2 geriatrics-09-00088-t002:** Themes, sub-themes, and summaries.

Theme	Sub-Theme	Summary
1. Creating meaningful communities	1.a. Meaningful relationships	Passengers form bonds with pilots and other passengers, enjoying conversations and social experiences on the rides.
	1.b. Shared experience	Bike rides foster conversation and comfortable quiet moments, enhancing social connections and providing opportunities to share experiences with others during rides and beyond.
2. Breaking the monotony of everyday life		Bike rides offer a refreshing break from the routine of nursing home life, providing new experiences, nature interactions, and a sense of renewal.
3. Reconnecting to oneself	3.a. To be seen and noticed	Passengers enjoy being noticed by others outside the nursing home, which creates a sense of connection to the outside world.
	3.b. To revisit and reminisce	Revisiting familiar places allows passengers to reminisce and reflect on their lives, providing deep emotional and existential significance.

## Data Availability

Because of privacy, the data presented in this study are available on request from the corresponding author.

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
