# Peer review of "Can We Steer Nursing Home Residents Away from Loneliness? A Qualitative Study of Cycling without Age"

_geriatrics, 2024, doi:10.3390/geriatrics9040088_

Round 1

Reviewer 1 Report

Comments and Suggestions for Authors

The articles is very interesting, even if the sample size is very small i would recommend some revisions.  

Introduction

Please, add the hypothesis at the end of the introduction and the added value of your work. 

Methods 

Did authors obtained informed consent? 

Who run the interview? 

Why did authors included only 8 patients? Did authors performed the power estimation of the sample? 

Did authors follow a theoretical framework to design the interview?

Results

I suggest to create a table in order to improve the reading of the results section 

Discussion 

Line 432 there is an extra space

Comments on the Quality of English Language

Minor editing of English language required

Author Response

Dear reviewer,

Thank you for your commentary. We greatly appreciate the feedback you provided.
Please find below a table with rows for your comment, a response from the authors, and a line/page number or otherwise to indicate revisions to the manuscript. Find the article with tacked change in the attached file, where reviewer 2 suggestion for change also are. 

Editorial/reviewer comments

Author response

Revision to manuscript

Reviewer 1

Please, add the hypothesis at the end of the introduction and the added value of your work. 

We have reviewed the introduction based on your suggestion.

Added to line 73-74:

Our hypothesis is that participating in CWA mitigates the experience of loneliness among nursing home residents.

Added to line 76-78

This study will add to providing evidence-based recommendations to enhance the quality of life for this vulnerable population.

Did authors obtain informed consent? 

Informed and written consent was obtained from all participants.

We have moved the following sentence further up in the method section (from line 150-151), to highlight that we obtain informed consent.

Line 118-119

We obtained informed and written consent from all participants. Pseudonyms for the passengers are used throughout the paper.  

In line 548-549 we also declare that we obtained informed consent.

Who run the interview?

Added initials of authors conducting interviews.

We have now included this information in the method section (Line 155):

Interviews were conducted by MHS and SKL

Why did authors included only 8 patients?  Did authors performed the power estimation of the sample?

No power calculation was conducted, as power calculations are typically associated with quantitative research to determine the sample size needed to detect an effect with a certain level of confidence. In qualitative research, such as this study, the focus is on depth and richness of data rather than statistical power; this in turn partially answers why no more than 8 nursing home residents were included.

We have added a flowchart at line 122

Did authors follow a theoretical framework to design the interview?

We followed Kallio et al’s theoretical framework to designing an interview.

We have added the following sentence at line 157-162:

To develop our semi structured interview guide we used the framework by Kallio et al. [27] In the development, we went through the following steps: (1) identifying the prerequisites for using semi-structured interviews, (2) retrieval and use of previous knowledge, (3) formulating the preliminary semi-structured interview guide, (4) pilot testing the interview guide, (5) presenting the complete semi-structured interview. (appendix 1)

We have further included both interview guide and observation guide in the appendix (appendix 1 and appendix 2).

I suggest to create a table in order to improve the reading of the results section 

We have now inserted a more elaborate table 2 that briefly summarizes the results to improve the reading of the results. We have also named the themes headlines with the same numbers and letters after the headlines in the table.

We have added a new version of Table 2 to line 223.

Line 432 there is an extra space

Removed an extra space from line 432.

Reviewer 2 Report

Comments and Suggestions for Authors

I commend the authors for their well-written study, which I believe succeeded in capturing the essence of the values and opinions of their cohort. The study describes the lived experiences of participants in the CWA program, a social initiative that takes nursing home residents out for a bike ride led by a volunteer pilot. A particular focus is given to the role of CWA in mitigating loneliness. Results from their interviews and observations highlight the importance of creating communities, breaking the monotony of daily life, and reconnecting with oneself, which are the main benefits listed by participants.

The main issue with the manuscript is that it is hard to find a critical note in the reporting. This might, of course, be because CWA is a great initiative (and from my perspective so far, I believe that this might be the case), but it could also be a design or reporting issue. For this reason, I would like the authors to make the interview or topic guide available as a supplementary file. Second, it would be good to highlight any critical responses or mention that there were none. For example, did anyone express sadness about being unable to paddle themselves or the short range on a bicycle? Last, it would be good to add how many of the nursing home residents, in relation to the total population, engage in CWA (this adds to the limitation of the biassed sample).

A critical note might also be added to the discussion section in the discussion about the limitations of implementing CWA on a large scale. In many countries around the world, we see a rise in the number of people who need care and a decrease in the people who can care for them. As such, how realistic is implementing CWA, which needs a 1 on 1 or 1 on 2 volunteer interaction? I understand that this is beyond the scope of the study, but I wonder whether you can discuss the question of what we should take away from this study with CWA that we need to retain also when we cannot reach these volunteer numbers?  

Other than these overarching points, please consider the next smaller issues:

Page 3, line 98: “Those who were unable to participate in a minimum 20-minute interview were excluded from the interview phase” How did you assess the ability to complete the interview? How much does this influence your statement about avoiding an elite sample, made later on this page (line 118)?

Page 3, line 108: Please mention that these are pseudonyms at the first listing of the names.

Page 3, table 1: How was cognitive status assessed?

Page 3, table 1: Please refrain from using the word ‘Normal’ when discussing cognitive status and consider using terms like ‘neurotypical’ or ‘no known disorders’.

Page 9, line 352: would it be clearer to phrase this as ‘… lack of independence as a resident of a nursing home’?

Page 11, line 455: The sentence “This study makes a valuable contribution by pointing out the great potential of CWA to mitigate loneliness” implies causality that cannot be concluded from this study. Perhaps CWA participants are more social and less lonely people to begin with. By limiting the sample to active CWA participants, the ability to infer relationships is very limited. 

Author Response

Dear reviewer,

Thank you for your commentary. We greatly appreciate the feedback you provided.
Please find below a table with rows for your comment, a response from the authors, and a line/page number or otherwise to indicate revisions to the manuscript. Find the article with tacked change in the attached file, where reviewer 1 suggestions for change also are. 

Editorial/reviewer comments

Author response

Revision to manuskript

Reviewer 1

The main issue with the manuscript is that it is hard to find a critical note in the reporting. This might, of course, be because CWA is a great initiative (and from my perspective so far, I believe that this might be the case), but it could also be a design or reporting issue. For this reason, I would like the authors to make the interview or topic guide available as a supplementary file.

Thank you for your commentary. We appreciate the angle you put forth. We will add both the observational and interview guides to the appendix.

To addressee your concern, which we appreciate, we want to highlight the question from our interview guide: Have you ever not wanted to go cycling? If so, how did you feel after going?” Where we give the interviewees an opening to address if there is a more negative aspect of CWA.

Otherwise we asked open-ended questions and followed up on any answers regardless of they were positive or negative.

Regarding your assessment of Cycling Without Age, an Australian study conducted around the same time as our data collection to place corroborates the idea of Cycling Without Age being a through and through great initiative: https://pubmed.ncbi.nlm.nih.gov/35864590/

To addressee your concern about that it is hard to find a critical note and our reporting about only positive perspective, we have added a section in the discussion, were we asses, that it may be a biassed sample since we only interviewed active CWA passengers, and didn’t talk with nursing home residents, who wasn’t active in CWA.

We have added the interview guide as appendix 1 and the observation guide as appendix 2

At line 496-499 we discuss our biassed sample further

In general, our results are based on informants who are active CWA passengers who perceive CWA positively. We may therefore lack diverse and critical perceptions and experiences, this may due to our sample is some degree of biassed by only including passengers form CWA.

Second, it would be good to highlight any critical responses or mention that there were none. For example, did anyone express sadness about being unable to paddle themselves or the short range on a bicycle?

Thank you for addressing this. It is true we haven’t included any critical responses. That is because we didn’t find any thematic themes even though we asked about they have ever not wanted to go cycling.

We have added a part in the results, where we report that we didn’t find any critical themes.

At the same time we have added a part in the discussion, where we address our biassed sample (as mention in the answer above)

Included the following in the results at line 213-220:

The three themes do not reflect any negative association or bad experiences. All passengers were asked open-ended questions about their experiences with the CWA, including any negative aspects. While some passengers expressed that they preferred not to go on rides if the weather was too cold or rainy, that it could feel cramped to be two passengers in one bike and that they were dependent on a pilot to offer them a ride. However, despite great attempts to ask about and include any bad or negative sides of participating in CWA all passengers were overwhelmingly positive great full for the rides and that is what will be reflected in the three themes.

Last, it would be good to add how many of the nursing home residents, in relation to the total population, engage in CWA (this adds to the limitation of the biassed sample).

We have added a flowchart depicting the total number of residents living at the nursing homes we conducted our study at, (unknown how many of these actually go on rides!), how many were deemed eligible for interviews by staff at the homes, and how many we ended up interviewing.

Added a flowchart at line 122

A critical note might also be added to the discussion section in the discussion about the limitations of implementing CWA on a large scale. In many countries around the world, we see a rise in the number of people who need care and a decrease in the people who can care for them. As such, how realistic is implementing CWA, which needs a 1 on 1 or 1 on 2 volunteer interaction? I understand that this is beyond the scope of the study, but I wonder whether you can discuss the question of what we should take away from this study with CWA that we need to retain also when we cannot reach these volunteer numbers?

Thank you for this keen observation. We believe it might not be the cycling 1:1 or 1:2 with a volunteer in and of itself that makes the impact, rather, it is elements such as being confronted with objects or places from the past, allowing/forcing the residents to reminisce, as well as having new experiences in a late stage of life. And we believe this can take many shapes; Cycling without age being just one. 

At the same time we understand that it may be difficult to upscale CWA due to the rise in the number of people who need care as mention in the article from Spann et al. wich we have added to our references.

[34] Spann et al. Understanding the care and support needs of older people: a scoping review and categorisation using the WHO international classification of functioning, disability and health framework (ICF). BMC Geriatr 19, 195 (2019)

Added to line 474-481

However, implementing of CWA on a large scale is challenging due to the in-creasing elderly population, the increase in the need for care and decreasing number of people available to provide this care [34]. The volunteer model may not be realistic globally if. However, as the number og elderly rises, more elderly individuals become available after pension, and could be serving as volunteers themselves. Thus, even with limited volunteers, the benefits of CWA could be achieved by making it a regular part of care or by integrating CWA activities into nursing home routines that can provide meaningful experiences through resident interactions, outdoor activities, and community connections

Page 3, line 98: “Those who were unable to participate in a minimum 20-minute interview were excluded from the interview phase” How did you assess the ability to complete the interview?

Thank you for addressing this. We assess the ability to complete the interview in cooperation with the gatekeepers who know in advance if a passenger may be able to complete the interview. Before an interview, we talked with the informant ourselves to see if they were able to have a longer conversation

Added to line 102-105:

Eligibility was assessed mainly by the gatekeepers, in corporations with the project  team. Passengers who were unable to participate in a minimum 20-minute interview were excluded from the interview phase but were still considered for the observational aspect of the study.

How much does this influence your statement about avoiding an elite sample, made later on this page (line 118)?

There is no doubt that our interview sample is among the ‘elite’ of nursing home residents. This is the case for two reasons:
1) In order to obtain relevant, first-hand data, participants should not be demented, and they should have a verbal language.

2) In order to oblige to ethical guidelines, participants had to be able to give informed consent.

Our observations, however, included non-verbal residents, and thus provide us with data from non-elite residents.

 Added to line 138-143

Even though we included passengers with cognitive and communicative limitations we still had to exclude passengers who were unable to participate in a 20-minute interview leaving us with somewhat of an elite sample without demented participations able to have a verbal language, in order to obtain relevant, first-hand data and in order to ob-ligate ethical guidelines. Our observations, however, included non-verbal passengers, an this provided us with data from non-elite residents.

Page 3, line 108: Please mention that these are pseudonyms at the first listing of the names.

Thank you for pointing this out.

Moved the following to line 119-120:

We obtained informed and written consent from all participants. Pseudonyms for the passengers are used throughout the paper.“

Page 3, table 1: How was cognitive status assessed?

Thank you for spotting this.

We added to table 1 that the information was provided to us by the nursing home staff during the recruitment phase (line 155).

* Information about cognitive status was provided by the nursing home staff during the recruitment phase

Page 3, table 1: Please refrain from using the word ‘Normal’ when discussing cognitive status and consider using terms like ‘neurotypical’ or ‘no known disorders’.

Mention of “normal” cognitive status have been addressed.

Changed the use of ‘normal’ to ‘no known disorders’ in table 1. (line 155)

Page 9, line 352: would it be clearer to phrase this as ‘… lack of independence as a resident of a nursing home’?

Thank you for pointing this out.

Rephrased line 400

Page 11, line 455: The sentence “This study makes a valuable contribution by pointing out the great potential of CWA to mitigate loneliness” implies causality that cannot be concluded from this study. Perhaps CWA participants are more social and less lonely people to begin with. By limiting the sample to active CWA participants, the ability to infer relationships is very limited. 

We agree that the ambiguity of “might” is relevant.

Further we have added, your observation on the fact that our participants might be a biassed group on characteristics such as being social and less lonely

Added “might” at line 518-501, and added in the following sentence at the end of the paragraph line 519-521:

However, the sample is limited to the sample is active CWA participants, who may inherently be more social and less lonely to begin with and to compared to the general population.

Round 2

Reviewer 1 Report

Comments and Suggestions for Authors

Dear authors,

After this revision i think that the manuscript was improved. The  main limitation consist in the small sample size, even if we consider the study design. However, the authors explained and took into account this limitation in the discussion section.

Comments on the Quality of English Language

Minor english check are required. 

Author Response

Dear reviewer,

Thank you for your commentary. We greatly appreciate the feedback you provided.
Please find below a table with rows for your comment, a response from the authors, and a line/page number or otherwise to indicate revisions to the manuscript. Find the article with tacked language change in the attached file, where reviewer 2 suggestions for change also are. 

Editorial/reviewer comments

Author response

Revision to manuscript

Reviewer 1

Minor english check are required

Dear reviewer. Thank you for your feedback. We have gone through the article and reviewed the language. We have made some corrections. If you are not satisfied with the language, we can send it for linguistics proofreading. If you still deem it necessary, it will be returned to us on 17.06.2024.  

You can see the language corrections we made as ‘tracked changes.’ We have also uploaded a new version of the manuscript with accepted language corrections under "Submit Revised Manuscript". 

Reviewer 2 Report

Comments and Suggestions for Authors

The authors have done well in improving the manuscript. The issues around the objectivity of the study (e.g. the use of a biased sample and the absence of critical responses) have been remedied by adding more transparency. However, some issues remain, the main one is in the hypothesis that has been added to line 73-74, as a well-phrased hypothesis is vital for the scientific soundness of the manuscript. 

When a hypothesis is proposed, a conclusion is also needed in the discussion/conclusion section. This follow up at the end of the manuscript is currently missing. Secondly, this study is not suited to answer this hypothesis, as the hypothesis implies causality that cannot be tested without the use of an experimental design.

Line 101. “To be eligible, passengers had to resident a Danish nursing home affiliated with the CWA community” – please change ‘resident’ to ‘reside in’

Line 790-795. Some errors in spelling/sentence structure in the added part. Please make appropriate edits.

Line 827 – 829. This sentence does not flow well; please rephrase, for instance, by replacing ‘to’ with ‘as it’. 

Comments on the Quality of English Language

Some of the added sentences need to be reworked a bit.

Author Response

Dear reviewer,

Thank you for your commentary. We greatly appreciate the feedback you provided.
Please find below a table with rows for your comments, a response from the authors, and line/page numbers or other indications of revisions to the manuscript. Find the manuscript with tracked changes in the attached file, where you and Reviewer 1's suggestions for an English check are included (document with language corrections). You can see the language corrections we made as ‘tracked changes,’ but as the document seems a bit overwhelming, we have also uploaded a new version of the manuscript with accepted language corrections under 'Submit Revised Manuscript' (document with accepted language corrections)." If you or reviewer 1 are not satisfied with the language, we can send it for linguistics proofreading. If you still deem it necessary, it will be returned to us on 17.06.2024.  

Editorial/reviewer comments

Author response

Revision to manuskript

Reviewer 1

The authors have done well in improving the manuscript. The issues around the objectivity of the study (e.g. the use of a biased sample and the absence of critical responses) have been remedied by adding more transparency. However, some issues remain;

The main one is in the hypothesis that has been added to line 73-74, as a well-phrased hypothesis is vital for the scientific soundness of the manuscript.

When a hypothesis is proposed, a conclusion is also needed in the discussion/conclusion section. This follow up at the end of the manuscript is currently missing. Secondly, this study is not suited to answer this hypothesis, as the hypothesis implies causality that cannot be tested without the use of an experimental design

Thank you for pointing this out. We agree with you and have rephrased the hypothesis in a way that is more in lign with our qualitative design.  

Rephrased from

"Our hypothesis is that participating in CWA mitigate the experience of loneliness among nursing home residents that participate in CWA.

We aimed to explore nursing home residents' lived experiences of the rides and whether participation in CWA can mitigate experiences of loneliness. This study will add to providing evidence-based recommendations to enhance the quality of life for this vulnerable population.”

To:

“This study explores the experiences of nursing home residents who participate in Cycling Without Age (CWA) rides, with a particular focus on their perceptions of loneliness. It aims to understand the residents' lived experiences of the rides and how they influence feelings of loneliness. The findings from this study will contribute to evidence-based recommendations for enhancing the quality of life for this vulnerable population.”

Line 83-87 in the document with language corrections and line 70-74 in the document with accepted language corrections.

Line 101. “To be eligible, passengers had to resident a Danish nursing home affiliated with the CWA community” – please change ‘resident’ to ‘reside in’

Thank you for noticing this. We have now changed it in accordance with your suggestion.

Line 112-113 in the document with language corrections and line 95-96 in the document with accepted language corrections

Line 790-795. Some errors in spelling/sentence structure in the added part. Please make appropriate edits.

We appreciate that you brought this to our attention. Unfortunately, there seems to be some mix-up in the line numbers. We have gone through our article thoroughly and hopefully found the phrases you were referring to. We have changed it.

We have looked thought the mention line and changed the spelling.  Line 509-517 in the document with language corrections and line 450-458 in the document with accepted language corrections

Old sentence “However, implementing CWA on a larger scale might be challenging due to the increasing elderly population, the increase in the need for care and decreasing number of people available to provide this care [34]. The volunteer model may not be realistic globally if. However, as the number of elderly rises, more elderly individuals become available after pension, and could be serving as volunteers themselves. Thus, even with limited volunteers, the benefits of CWA could be achieved by making it a regular part of care or by integrating CWA activities into nursing home routines that can provide meaningful experiences through resident interactions, outdoor activities, and community connections.”

New sentence

“However, implementing Cycling Without Age (CWA) on a larger scale may present challenges due to the increasing elderly population, the rising demand for care, and the decreasing number of available caregivers [34]. Moreover, the volunteer model of CWA may not be globally feasible. Nonetheless, as the population of elderly grows, more individuals who have retired may become available to serve as volunteers. Therefore, despite limited volunteer availability, the benefits of CWA could still be realized by making it a regular part of care or by integrating CWA activities into nursing home routines. This integration can provide meaningful experiences through resident inter-actions, outdoor activities, and community connections.”

Line 827 – 829. This sentence does not flow well; please rephrase, for instance, by replacing ‘to’ with ‘as it’.

Thank you. We have rephrased this part.

Line 520 in the document with language corrections and line 461-468 in the document with accepted language corrections, are chanced from

“This study provides an important contribution by pointing out the potential of CWA to might mitigate loneliness, and, in particular, the highly prevalent existential loneliness among nursing home residents. However, the sample is limited to the sample is active CWA participants, who may inherently be more social and less lonely to begin with and to compared to the general population”

To:

This study gives valuable insight into the importance of how an activity as CWA can affect nursing home residents´ perception of loneliness. A main strength of our study is that data comprise both observations and interviews, which increased our ability to capture nuances in the views of the nursing homes residents. Our observational notes gave valuable insight into the setting and helped us understand the experience of a ride. Additionally, the observations gave a voice to the nursing home residents who had difficulty expressing themselves verbally. We included informants from four different nursing homes to examine common features of rides in different contexts.
